# Clinical Correlates of Measured and Predicted Resting Energy Expenditure in Patients with Anorexia Nervosa: A Retrospective Cohort Study

**DOI:** 10.3390/nu14132727

**Published:** 2022-06-30

**Authors:** Rami Bou Khalil, Ariane Sultan, Maude Seneque, Sami Richa, Patrick Lefebvre, Eric Renard, Philippe Courtet, Laurent Maimoun, Sebastien Guillaume

**Affiliations:** 1Department of Psychiatry, Saint Joseph University-Hôtel Dieu de France Hospital, Mar Mikhael, Beirut 17-5208, Lebanon; sami.richa@usj.edu.lb; 2PSNREC, University of Montpellier, INSERM, CHU de Montpellier, 34295 Montpellier, France; maude.seneque@gmail.com (M.S.); p-courtet@chu-montpellier.fr (P.C.); s-guillaume@chu-montpellier.fr (S.G.); 3UMR CNRS 5203, Institute of Functional Genomics, University of Montpellier, INSERM U1191, 34295 Montpellier, France; a-sultan@chu-montpellier.fr; 4Department of Psychiatric Emergency and Acute Care, Lapeyronie Hospital, CHRU, 34295 Montpellier, France; 5Department of Endocrinology, Diabetes and Nutrition, CHRU, 34295 Montpellier, France; p-lefebvre@chu-montpellier.fr (P.L.); e-renard@chu-montpellier.fr (E.R.); l-maimoun@chu-montpellier.fr (L.M.); 6Institute of Functional Genomics, University of Montpellier, INSERM, CNRS, 34295 Montpellier, France; 7Département de Médecine Nucléaire, Hôpital Lapeyronie, Centre Hospitalier Régional Universitaire (CHRU) Montpellier, 34295 Montpellier, France

**Keywords:** anorexia nervosa, resting energy expenditure, AN duration, AN subtype

## Abstract

Resting energy expenditure (REE; i.e., the calorie amount required for 24 h during a non-active period) is an important parameter in nutritional rehabilitation of patients with anorexia nervosa (AN). This study determined whether age, body mass index, AN duration/subtype/specific symptoms/clinical severity, cognitive function alterations, and psychiatric comorbidities influenced REE or the difference between the calculated and estimated REE. Patients with AN who were followed at a daycare treatment facility between May 2017 and January 2020 (*n* = 138) underwent a complete assessment that included the MINI, Eating Disorder Examination Questionnaire, d2 test of attention, body fat composition by bioelectrical impedance analysis (BIA) and REE measurement by indirect calorimetry (REE_IC_). AN subtype (N = 66 for restrictive subtype and N = 69 for non-restrictive subtype; *p* = 0.005), free-fat mass (<0.001), and fat mass (<0.001) were associated with REE_IC_. Age (*p* < 0.001), height (*p* = 0.003), and AN duration (N = 46 for <3 years and N = 82 for ≥3 years; *p* = 0.012) were associated with the difference between estimated REE (using the Schebendach equation) and measured REE_IC_. Therefore, the Schebendach equation was adjusted differently in the two patients’ subgroups (AN duration ≤ or >3 years). Overall, REE was higher in patients with restrictive than non-restrictive AN. In the absence of BIA measures, REE-estimating equations should take into account AN duration.

## 1. Introduction

Anorexia nervosa (AN) affects 0.5–1% of women during their lifetime and approximately one tenth as many men [1]. The main AN features are intense fear of weight gain, food intake restriction, and body-image disturbances [2]. Due to its high social and professional impact and elevated morbidity and mortality, AN is one of the most severe mental disorders [3,4,5]. As AN is a multifactorial disorder, a multimodal treatment plan, which involves also nutritional rehabilitation [6], is required for its management.

In the available nutritional rehabilitation programs for AN, the calorie levels prescribed to promote weight restoration are often based on resting energy expenditure (REE) estimates. REE represents the amount of calories required by the body for 24 h during a non-active period. Indirect calorimetry techniques (e.g., metabolic cart and Douglas bag) are the gold standard method for measuring REE, but they are difficult to apply in standard clinical and hospital settings [7,8,9]. Several other computerized techniques are more practical to use in such settings, but they may present a variable margin of error due to the technology heterogeneity [8,9]. Therefore, predictive formulae to calculate REE, based on age, height, and body weight, such as the Harris–Benedict equation (elaborated for patients with malnutrition [6,10]), the Schebendach equation (an adaptation of the Harris–Benedict equation for patients with AN) [11], and the Muller equation [an adaptation of the Harris–Benedict equation for individuals with a body-mass index (BMI) <18.5] are widely used by clinicians [12]. Although their accuracy in estimating REE in patients with AN is debated [13,14], the level of agreement between indirect calorimetry measurements and the Muller equation, which includes fat-free mass (FFM) and fat mass (FM), is considered adequate [15,16]. FFM and FM can be determined by bioelectrical impedance analysis (BIA) or dual energy X-ray absorptiometry. BIA relies on formulae that use the measured reactance and resistance to estimate FFM and FM in different population settings [17,18]. Conversely, dual energy X-ray absorptiometry directly measures the body composition, but it is an expensive technique that sometimes is not available in clinical settings [19]. The level of agreement between BIA and dual energy X-ray absorptiometry findings is debated [20,21].

In addition, although many alterations are similar in patients with AN and in patients with malnutrition due to other causes, several AN-specific nutritional aspects, such as some behavior patterns, food selectivity, neuroendocrine changes, and clinical comorbidities, may increase the difference between predicted and measured REE [6,22]. This discrepancy between predicted and measured REE values may partly explain why in some patients with AN, nutritional rehabilitation fails (i.e., lack of weight restoration if the estimated REE is lower than the true REE, or refeeding syndrome if the estimated REE is higher than true REE [23]). The absolute REE is lower in underweight patients with AN than in controls. It increases during refeeding and is positively correlated with total body weight and FFM, partly because of muscle mass mobilization [24]. Hyperactivity is very frequent in patients with AN and is considered a metabolically expensive process that should lead to higher REE compared to controls or low-exercising patients, even after adjusting for lean body mass and body surface area [20]. However, hyperactivity or belonging to a specific AN subtype (e.g., restrictive or binge/purging subtype) does not seem to affect the REE [25]. Other factors (e.g., anxiety, depressive mood, nicotine consumption, abdominal pain) may influence REE in patients with AN, thus hindering weight restoration during refeeding [26]. The psychopathology severity in patients with AN may contribute to increase REE [27]. Biological factors, such as thyroid function, circulating cytokines and systemic cortisol levels, also may affect REE [28,29,30].

Moreover, AN is often considered to be an illness with different severity levels that might be expressed on a continuum [31]. It has been suggested that clinical staging could be a mean to simultaneously approach divergences in categories and dimensions [31]. One of the most important factors that might affect AN clinical staging is the disease duration because of the high instability of the AN subtype in the early phases [32]. For instance, the transition from restrictive to non-restrictive type is most likely to occur during the first year and the fifth year of disease with the median to onset of binge eating in restricting type AN being 24 months from the onset of illness [33]. In addition, the impact of the sustained hypercortisolism and appetite hormone dysregulation and the disease effect on adipose tissue and muscle mass are different in acute and chronic AN [34,35]. Besides these factors, chronic starvation and other behavioral phenotypes, such as physical hyperactivity and induced vomiting, may differentially alter body composition, with consequences on several important clinical, biological, and prognostic parameters, including REE [30,36,37]. However, it is not known whether AN duration influences REE.

To the best of our knowledge, no study simultaneously assessed the association of several possible AN categories with REE variability. In addition, all the available equations to estimate REE take into account only demographic, anthropometric, and body composition data (age, sex, height, weight, FFM, and FM) without considering that some specific AN clinical features also may be important. We hypothesized that some clinical features (e.g., AN duration and/or subtype, comorbid mental disorders, specific accompanying symptoms, cognitive function alterations, and clinical severity) may affect REE. The main objective of this study was to identify AN-related clinical variables and/or comorbidities that affect REE measured by indirect calorimetry and/or are associated with higher differences between predicted and measured REE. The secondary objective was to propose a modified equation for REE estimation that takes into account these significant variables and that is in agreement with the measured REE and easy-to-determine in clinical settings.

## 2. Materials and Methods

### 2.1. Participants

Eligible patients were all consecutive outpatients with AN, according to the Diagnostic and Statistical Manual of Mental Disorders 5th edition (DSM-5) criteria, who were seen at the eating disorder unit of Montpellier, France, between May 2017 and January 2020 [38]. Patients with AN are referred to this unit for multidisciplinary assessment, diagnosis confirmation, and management. The data used for this analysis were from a large study approved by the Clermont Ferrand University CPP Sud-Est VI Ethics Committee (CPP: AU 1313; ID-RCB: 2017-A00269-44; Clinical Trial number: NCT03160443). Signed informed consent was obtained from all participants (and from the parents of underage participants). All research procedures were performed according to the Declaration of Helsinki. Inclusion criteria were: (1) speaking French, (2) being a woman, and (3) AN diagnosis according to the DSM-5 criteria. Exclusion criteria were: (1) consent refusal, (2) mental disability, such as intellectual deficiency or psychotic disorder, (3) physical comorbidity that prevented the participation in the study, and (4) difference between estimated and measured REE of |±1000| kcal or higher. Among the 152 eligible participants, 138 were included in the study sample for statistical analyses.

### 2.2. Measures

The multidisciplinary clinical assessment was carried out at the outpatient unit by experienced mental health professionals during one whole day. AN diagnosis was established on the basis of a non-structured clinical assessment by psychiatrists, psychologists, and nutritionists, and of a structured evaluation with the Mini-International Neuropsychiatric Interview (MINI, version 5.0.0). All investigators were trained to use the MINI. Body weight and height were collected in a standardized manner during the clinical examination. The presence of specific AN symptoms (food selectivity, meal skipping, bulimic episodes, vomiting, physical hyperactivity, snack eating, compulsory eating, polydipsia, fasting practices, and laxative use) was assessed using close-ended questions.

All participants completed the following questionnaires:

**MINI:** this is a structured evaluation of psychiatric disorders according to the DSM-5. The interview is carried out by a trained mental health professional and is based on multiple branching flowcharts that lead to a specific diagnosis if several criteria are met. The aim of the MINI interview was to assess the presence of one of the frequently encountered psychiatric comorbidities in patients with AN (depressive disorder, anxiety disorder, obsessive compulsive disorder, post-traumatic stress disorder, alcohol and/or other substance use disorder) and/or a history of suicide attempt.

**Eating disorder examination questionnaire (EDE-Q):** this 28-item self-report questionnaire is used to evaluate eating disorder symptoms over a 28-day period. It gives a total score, and scores for four subscales that explore the four core clinical dimensions of eating disorders: restraint, eating concern, body shape concern, weight concern [39]. Higher total and subscale scores indicate more problematic eating behaviors and attitudes. For this study, the total score was defined as a continuous variable for bivariate comparisons [40]. The Cronbach’s alpha coefficients for the subscale scores were: 0.857 for restraint, 0.762 for eating concern, 0.831 for weight concern, and 0.913 for shape concern. The Cronbach’s alpha coefficient for the total score was 0.812.

**D2 test of attention:** in this paper–and–pencil cancellation test, participants must find target items (d2) among distracter items in a series of consecutively ordered items (14 rows with 47 characters/each for a total of 658 items). Participants are instructed to cancel as many target symbols as possible, by moving on the page in a reading-like manner from left to right with a time limit of 20 s per trial without break. The ratio between the number of misses (omission errors: number of “d” with two marks that were not marked) and the total number of items processed was the main variable considered in this study [41].

In addition, REE and body fat composition were assessed in all patients. BIA was used to estimate FFM and FM. Resistance and reactance were determined using a four-terminal impedance plethysmograph (BIACORPUS RX 4000, Lainate, Italy). REE was determined using open-circuit indirect calorimetry (REE_IC_) with the QUARK RMR system (COSMED, Rome, Italy). Measurements were obtained in the morning, between 8 a.m. and 10 p.m., after completing overnight fast and smoking cessation. REE was estimated using the following equations:Harris–Benedict equation: REE_HB_ (Kcal/day) = 655.1 + (9.56 × weight in kg) + (1.84 × height in cm) − (4.67 × age in years) [10]Schebendach equation: REE_Sch_ (Kcal/day) = (1.84 × Harris-Benedict predicted REE) − 1435 [12]Muller 1 equation: REE_Muller1_ (Kcal/day) = 239 × [0.07122 × weight in kg − 0.02149 × age in years + 0.731] [11]Muller 2 equation: REE_Muller2_ (Kcal/day) = 239 [0.08961 × FFM in kg +0.05662 × FM in kg + 0.667] [11].

### 2.3. Statistical Analyses

Quantitative variables that significantly departed from the normality assumptions (assessed with the Shapiro–Wilk test) were expressed as medians with interquartile ranges (IQR: Q1–Q3). Otherwise, variables were described as mean ± standard deviation (SD). All variables, except for the measured and estimated REE, FM, FFM, and height, were not normally distributed.

First, REE_IC_ was compared among groups created in function of categorical variables (AN subtype, food selectivity, meal skipping, bulimic episodes, vomiting, physical hyperactivity, snack eating, compulsory eating, polydipsia, fasting practices, laxative use, nicotine consumption, current depressive episode, current anxiety disorder, current obsessive compulsive disorder, current post-traumatic stress disorder, current alcohol and substance use disorder, history of suicide attempt). It was then correlated with continuous variables (Age, current BMI, current weight, current height, FFM, FM, AN duration, EDE-Q restriction subscale, EDE-Q eating concern subscale, EDE-Q weight concern subscale, EDE-Q shape concern subscale, EDE-Q total score, d2 score ratio) using the Pearson’s correlation or Spearman’s Rho correlation as non-parametric test. The Student’s *t* test (or the Mann–Whitney U non-parametric test) was used for the bivariate comparison of characteristics among groups. Nominal variables with small size groups (N < 30) were grouped, in accordance to the scientific rationale, by merging categories. The Cronbach’s alpha values were computed for the EDE-Q total and subscale scores. Then, REE_IC_ was considered as the dependent variable in a linear regression model. Independent variables were included in the final regression model if they were associated with REE_IC_ variability in the bivariate analysis (*p* < 0.15). The included continuous independent variables were classified, in function of their distribution quartiles, into categories that respected their original distribution. Independent variables were all added and excluded from the regression model in a backward fashion starting with the variables with the highest *p* value, while making sure that the adjusted R^2^ of the ANOVA test increased and the variance inflation factor (VIF) remained <1.2. On the basis of the degree of freedom (*df*), independent variables with >10% of missing values were excluded. In Model 1, all eligible independent variables were included (FM, FFM, AN duration in quartiles, AN subtype, food selectivity, meal skipping, compulsory eating, snack eating, bulimic episodes, vomiting, nicotine consumption, and history of suicide attempts). As BMI, height and weight were highly correlated with FM and FFM, they were not included in Model 1. In Model 2, only independent variables with the lowest *p* value and explaining the REE_IC_ variability remained (FM, FFM, AN duration, AN subtype, food selectivity, and nicotine consumption).

The level of agreement between REE_IC_ and the REE values estimated using the four equations was assessed. First, the difference between predicted and measured mean REE values [i.e., Diff_(HB-IC)_, Diff_(Sch-IC)_, Diff_(Muller1-IC),_ and Diff_(Muller2-IC)_] was tested with the one-sample *T* test to compare the difference when different from zero. Second, the level of agreement was checked visually according to the Bland–Altman method [42]. Briefly, the plots with the Diff_(HB-IC)_, Diff_(Sch-IC)_, Diff_(Muller1-IC)_, and Diff_(Muller2-IC)_ on the Y axis and the mean values of the corresponding estimated REE and REE_IC_ on the X axis were visually examined. For the estimated REE that was most in agreement with REE_IC_, the plot should show a mean difference close to zero and a narrow 95% confidence interval (CI) [mean ± 1.96 × SD].

The REE-estimating equation with the highest level of agreement with REE_IC_ and with the most practical applicability (which is the Schebendach’s equation) was then adjusted to improve REE prediction. First, in a bivariate analysis, the difference between the estimated and measured REE values [Diff_(Sch-IC)_] was correlated with continuous variables using the Pearson’s correlation or Spearman’s Rho correlation as non-parametric test (Age, current BMI, current weight, current height, FFM, FM, AN duration, EDE-Q restriction subscale, EDE-Q eating concern subscale, EDE-Q weight concern subscale, EDE-Q shape concern subscale, EDE-Q total score, d2 score ratio). The Student’s *t* test or Mann–Whitney U non-parametric test was used for the bivariate comparison of characteristics among groups (AN subtype, food selectivity, meal skipping, bulimic episodes, vomiting, physical hyperactivity, snack eating, compulsory eating, polydipsia, fasting practices, laxative use, nicotine consumption, current depressive episode, current anxiety disorder, current obsessive compulsive disorder, current post-traumatic stress disorder, current alcohol and substance use disorder, history of suicide attempt). Nominal variables with small size groups (N < 30) were grouped by merging categories. After having examined the correlation plot of Diff_(Sch-IC)_ and AN duration, the latter was dichotomized according to the comparison of the effect size, according to the Cohen’s d, of different cut-offs (2, 3, 4, 5, and 6 years) that were selected in function of published data on AN clinical staging [31,32,33]. In the multivariate analysis, the difference between predicted and measured REE values [Diff_(Sch-IC)_] was considered as the dependent variable. In Model 3, all eligible independent variables were included (age, current BMI, current height, AN duration in two categories, AN subtype, food selectivity, meal skipping, bulimic episodes, vomiting, snack eating, compulsory eating, fasting, laxative use, nicotine consumption, and history of suicide attempts). As current weight, FFM and FM were highly correlated with BMI and height, and they were excluded from the model. In Model 4, only independent variables with the lowest *p* value and better explaining Diff_(Sch-IC)_ variability remained (age, AN duration, AN subtype, current height, compulsory eating, food selectivity, fasting, nicotine consumption).

In the adjustment of the equation that gave the REE value most in agreement with the REE_IC_ value, AN duration was more relevant as categorical variable. Therefore, two equations were suggested for each subgroup of patients divided according to AN duration. The equation adjustment was done in two steps. In the first step, a regression analysis with the difference between predicted and measured REE values as independent variable was performed in each subgroup. A retained predictor (x) should increase the adjusted R^2^. In each subgroup, the corrected equation was:

**Difference between predicted and measured REE = a(x) + b**,

where a is the coefficient Beta of the predictor, and b is the value of y when all predictors are equal to zero.

The predicted REE was then calculated as follows:


**Predicted REE = Predicted REE according to the equation most in agreement with REE_IC_-a(x) − b.**


In the second step, the mean difference between predicted and measured REE values was determined and added to all estimated REE values in order to reduce the estimation error and bring the residuals of the regression equation closest to zero using the following equation:


**Predicted REE = Predicted REE according to the equation most in agreement with REE_IC_ + Mean difference between predicted and measured REE**


Lastly, a repeated measures ANOVA was used to compare REE_IC_, REE_HB_, REE_Sch_, REE_Muller1_, REE_Muller2_, and adjusted REE_Sch_, and a post-hoc analysis with Bonferroni correction was used to compare REE_IC_ with every other estimated REE.

All statistical analyses were performed using JASP 0.14.1.0 and SPSS 25.0.

## 3. Results

### 3.1. Sample Description

The median age of the 138 participants was 22 (19–28.75) years and their median BMI at the time of assessment was 17.68 (16.05–19.46) kg/m^2^. The median AN duration was 5 (2–11) years, and AN was non-restrictive in 50% of participants. Nicotine consumption was reported by 34.78% of participants. Anxiety disorder was the most frequent AN comorbidity (47.1%), followed by depressive episode (25.36%) and history of attempted suicide (20.29%). The mean REE_IC_ was 1135.65 ± 202.2 Kcal/day. The median d2 ratio was 0.032 (0.017–0.056) (Table 1).

### 3.2. Analysis of Factors Associated with REE_IC_

REE_IC_ was correlated with BMI, weight, height, FFM, and FM (Spearman’s Rho = 0.657 and *p* < 0.001, Spearman’s Rho = 0.689 and *p* < 0.001, Pearson’s correlation coefficient = 0.253 and *p* = 0.004, Pearson’s correlation coefficient = 0.595 and *p* < 0.001, Spearman’s Rho = 0.599 and *p* < 0.001, respectively) and also with AN duration (Spearman’s Rho = 0.203 and *p* = 0.021). REE_IC_ was significantly higher in patients with nicotine consumption (*p* = 0.03), history of suicide attempt (*p* = 0.004), bulimic episodes (*p* = 0.041), snack eating (*p* = 0.003), compulsory eating (*p* = 0.01), and non-restrictive AN subtype (*p* < 0.001). In the final linear regression analysis (Model 2), only AN subtype (*p* = 0.005), FFM (<0.001), and FM (<0.001) remained associated with REE_IC_ variability (Table 2).

### 3.3. Level of Agreement between Estimated REE and REE_IC_

The REE values estimated with the four equations (see Methods) were compared with the REE_IC_ value using the Bland–Altman method. The difference between the four predicted REE values and REE_IC_ was significantly different from zero according to the one-sample *T* test [Diff_(HB-IC)_: *t* = 11.32 and *p* < 0.001; Diff_(Sch-IC)_: *t* = −12.51 and *p* < 0.001; Diff_(Muller1-IC)_: *t* = −20.8 and *p* < 0.001; Diff_(Muller2-IC)_: *t* = −2.58 and *p* = 0.011]. According to the Bland-Altman graphs, REE_Muller2_ was the closest to REE_IC_ (mean difference of −31.96 Kcal, 95% CI: −309.08 Kcal–245.18 Kcal). However, the Muller 2 equation used to calculate REE_Muller2_ requires FFM and FM data obtained by BIA. In clinical settings where BIA might not be available, REE_Sch_ was the closest to REE_IC_ (mean difference of −183.24 Kcal, 95% CI: −511.6 Kcal–145.11 Kcal) (Figure 1).

### 3.4. Analysis of Factors Associated with Diff_(Sch-IC)_ Variability

Diff_(Sch-IC)_ was correlated with age, height, and AN duration (Spearman’s Rho = −0.497 and *p* < 0.001, Pearson’s correlation coefficient = 0.232 and *p* = 0.008, Spearman’s Rho = −0.572 and *p* < 0.001, respectively). Diff_(Sch-IC)_ was higher in patients with non-restrictive AN (*p* = 0.002), with nicotine consumption (*p* = 0.016), without food selectivity (*p* = 0.026), and with snack eating (*p* = 0.006). For the multivariate analysis, AN duration was categorized as an independent variable after testing several cut-off values (2, 3, 4, 5, and 6 years) and their Cohen’s d effect size (1.11, 1.13, 1.09, 1.12, and 0.99 respectively). Accordingly, the AN duration cut-off of 3 years was considered as having the best between-group differentiation (N = 46 and Diff_(Sch-IC)_ = −77.85 +/− 146.36 Kcal for <3 years vs. N = 82 and Diff_(Sch-IC)_ = −245.3 +/− 149.14 Kcal for ≥3 years; *p* < 0.001). Age (*p* < 0.001), height (*p* = 0.003), and AN duration (*p* = 0.012) were the independent variables significantly associated with Diff_(Sch-IC)_ variability (Model 4 in Table 3).

### 3.5. Adjustment of the Schebendach Equation to Estimate REE_Sch_ Using Factors Associated with Diff_(Sch-IC)_ Variability

Age, height, and AN duration were used to adjust the Schebendach equation. AN duration was used as a categorical variable to separate the study sample in two groups. Two linear regression models with Diff_(Sch-IC)_ as dependent variable were performed in each AN duration group. The new equation generated from each model included also age and height as factors that might influence Diff_(Sch-IC)_ variability. Height was an important predicting factor in the subgroup with AN duration ≤3 years (N = 47) and age in the subgroup with AN duration >3 years (N = 88). After correcting by adding the mean Diff_(Sch-IC)_ in each subgroup, the following corrected equations were obtained:

In the subgroup with AN duration ≤3 years:


**Predicted REE = (17.59 × weight in kg) + (3.017 × height in cm) − (8.59 × age in years) − 92.62**


In the subgroup with AN duration >3 years:


**Predicted REE = (17.59 × weight in kg) + (3.38 × height in cm) − (8.01 × age in years)**


### 3.6. Comparison between the Adjusted REE_Sch_ and REE_IC_

A repeated measures ANOVA showed that REE_IC_, REE_Sch_, REE_Muller1_, REE_Muller2_ and the adjusted REE_Sch_ were not equal (*p* < 0.001). In the post-hoc analysis, all estimated REE values were different from the REE_IC_ value except the adjusted REE_Sch_ (Table 4; Figure 2).

## 4. Discussion

To our knowledge, this is the first study that evaluated several clinical and/or psychopathological dimensions and categories known to be present in patients with AN with the aim of identifying factors associated with REE variability. Our findings showed that FFM, FM, and AN subtypes are associated with higher REE_IC_.

FFM and FM are altered in states of malnutrition [43]. In undernourished patients without AN, a negative energy balance is followed by a reduction in FFM, FM, and REE and the consequent increase in appetite to maintain homeostasis [43,44]. Changes in the metabolically active muscle mass (indirectly reflected by FFM variations) and a reduction in thermogenesis due to the FM decrease may be more prominent in the first phases of reduced food intake [43,44]. In patients with AN, FFM and FM variations also affect REE, but FFM is less metabolically active compared with other malnourished individuals [45]. This implies that in individuals with the same weight, REE might be lower in patients with AN than in other undernourished patients. Therefore, FFM and FM may be better predictors of REE than weight and BMI.

AN subtype is another important factor associated with REE_IC_ variability. In our study, REE_IC_ was lower in patients with restrictive than non-restrictive AN (1052.4 ± 177.56 vs. 1205.4 ± 184.96, *p* < 0.001), unlike what was reported by another study that compared 39 patients with restrictive and 23 patients with non-restrictive AN [26]. This discordance might be related to differences in the used methods. However, in patients with AN and bingeing and purging practices (as observed in non-restrictive AN), the negative energy balance effect is not as important as in patients with restrictive AN, and this might explain their higher REE_IC_. In agreement, patients with bulimic episodes (*p* = 0.04), snack eating (*p* = 0.003) and compulsory eating (*p* = 0.01) were the only subgroups with significantly higher REE_IC_ compared with all the other patients with AN. However, in the multivariate analysis, these eating-related behaviors did not remain associated with REE_IC_, unlike the AN subtype. This suggests that the AN clinical subtype must guide the nutrition rehabilitation plan rather than any other clinical category based on specific symptoms. On the basis of these findings, we hypothesize that for patients with non-restrictive AN, the daily caloric intake should be higher, independently of their FFM, FM, and BMI, because the muscle mass reaction to starvation and refeeding might be different according to the AN subtype. More studies are needed to test this hypothesis.

To our knowledge, our study is the first to propose that REE should be estimated in two steps and that AN duration should be taken into account when developing a REE prediction equation. Our findings showed that if FFM and FM cannot be measured, the REE estimated using the equation elaborated by Schebendach et al. as an adjustment of the Harris–Benedict equation for patients with AN was the most in agreement with the measured REE_IC_ [12]. The difference between estimated and measured REE values was influenced particularly by age (*p* < 0.001), height (*p* = 0.005), and AN duration (*p* = 0.018). Previous studies classically focused on age, weight, and body composition to adjust the accuracy of the available equations to estimate REE in individuals without AN [46,47,48]. However, to better predict REE in patients with AN, an estimating equation should include factors that best predict FFM and FM during the disease course. Studies that evaluated REE predicting equations in patients with AN did not report satisfactory results, especially when using equations that do not include body composition parameters [14,15,36,49]. Moreover, these studies did not determine what factors strongly influence the difference between measured and estimated REE. Our findings show that the classical REE estimation in patients with AN based on age, height, weight, and sex misses an important aspect of REE variability: the metabolic profile differences of patients at different disease stages. Besides the finding that the AN subtype seems to be an important factor when developing nutrition rehabilitation strategies, our data indicate that AN duration should be taken into account when estimating REE in the absence of body composition measurements. Accordingly, the Schebendach equation to estimate REE seems to perform better in patients with shorter (≤3 years) than longer (>3 years) AN duration (Diff_(Sch-IC)_: −77.88 ± 146.39 vs. −245.3 ± 149.14, *p* < 0.001). Moreover, in the subgroup of patients with shorter AN duration, height was more weighted as a predictor in the Schebendach and Harris–Benedict equations. This is in line with other studies showing the importance of height, besides weight, for predicting REE in adults [50]. However, as in our analysis FFM and FM were important REE_IC_ determinants, we think that in patients with shorter AN duration, height is not as important for REE estimation as the FFM and FM modifications observed during the first years of AN. This might partly be explained by the fact that AN usually starts during adolescence when height is still not stable, while the negative energy balance affects the muscle and other organs’ mass [36]. In the subgroup of patients with shorter AN duration, age was more weighted in the Schebendach equation. Age has always been implicated in predicting REE in humans, but it seems to have a lower role in the Schebendach equation for patients with longer AN duration. Evidence supports the fact that after AN onset, REE is suppressed as an adaptive mechanism to protect the muscle and other important organs’ mass [51]. However, in the long term, muscle might become chronically affected, bone density might decrease, and FM might strongly decrease [52,53,54]. Therefore, we could hypothesize that after few years of disease, a new energy balance develops in response to the changing body composition, and consequently AN duration influences less REE estimation. In addition, body composition is different in patients with restrictive and with non-restrictive AN [55]. As some patients with restrictive AN will transition to non-restrictive AN during the first 3 years of disease, differences in REE prediction might partly be related to differences between AN subtypes. Unfortunately, the analysis in which AN subtype was considered did not reach statistical significance (*p* = 0.113). Nevertheless, we think that this factor should be taken into consideration in future studies to develop equations to estimate REE in function of the AN stage.

Our study has several limitations. First, the assessment of some AN clinical characteristics, such as hyperactivity, was done using close-ended questions instead of objective measurements. Second, in our statistical analysis, some groups were very small in size (e.g., patients with current post-traumatic stress syndrome) or were merged with other groups to increase their size (e.g., patients with current substance and/or alcohol use disorder). This did not help to accurately determine the real effect of these variables on REE which limits the generalizability of these findings. However, none of the factors determining these groups (except for laxative use which was slightly associated with Diff_(Sch-IC)_ variability; *p* = 0.111) was statistically significant or tended toward significance in all our analyses which make our results sufficiently reliable. This does not preclude the fact that these factors, especially laxative use, warrant further consideration in studies with larger samples. Third, the adjusted REE_Sch_ equations were not validated in a different sample of patients with AN. Fourth, we have exclusively considered patients admitted to the daycare hospital of one university center at Montpellier, France, due to several logistic difficulties in recruiting patients from other centers and different settings. Accordingly, this also limits the generalizability of our findings to all patients with AN.

## 5. Conclusions

In our sample of women outpatients with AN, we found that FFM, FM, and AN subtypes are the most important factors associated with REE_IC_ variability. The REE values estimated using the Muller equation that included FFM and FM data were in agreement the most with REE_IC_. As FFM and FM measurements may not be widely available in general clinical settings, we adjusted the Schebendach equation by taking into account the factors that influence the difference between estimated REE_Sch_ and REE_IC_. Accordingly, based on the staging theory for AN classification and management, we adjusted the equation differently in function of AN duration (≤3 years and >3 years). The generalizability of our results might be limited mostly due to the monocentric nature of our study and the small group effect of some studied factors. Future studies should take into consideration AN subtype and duration in any REE-estimating equation. AN subtype should also be considered in any staging strategy because it affects REE and consequently AN management and prognosis.

## Figures and Tables

**Figure 1 nutrients-14-02727-f001:**
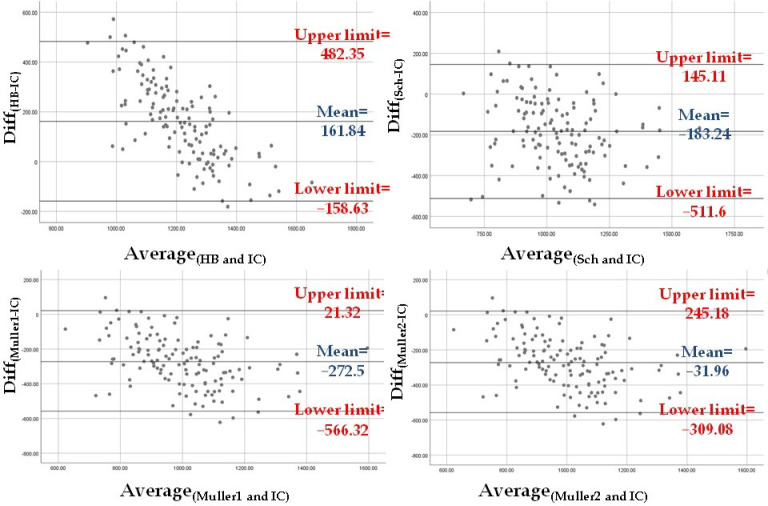
The Bland–Altman plots show that the REE estimated with the Muller equation that requires BIA measurements (Muller 2) is in agreement with REE_IC_ the most, followed by the REE estimated with the Schebendach equation.

**Figure 2 nutrients-14-02727-f002:**
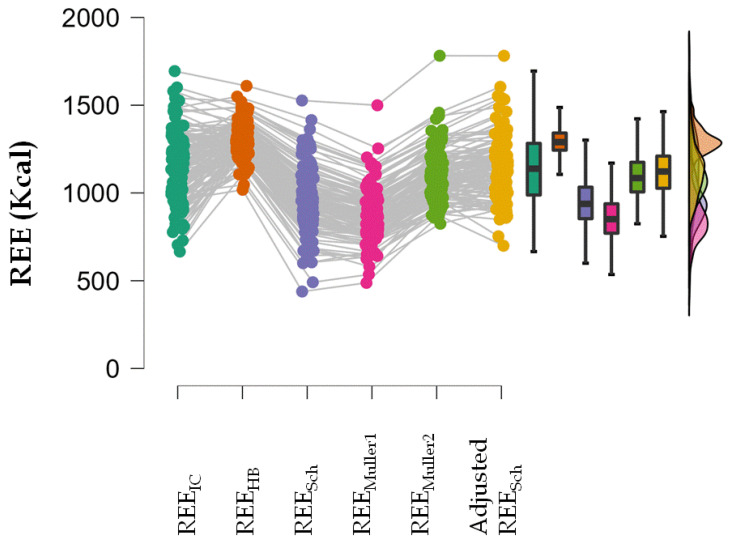
Raincloud plot of repeated measures ANOVA showing the distribution of the estimated REE and REE_IC_ values.

**Table 1 nutrients-14-02727-t001:** Descriptive analysis of the variables assessed in the study sample.

Variable	Mean +/− SD * [or Median (Q1–Q3)]	Category: N (%)
Age (years)	22 (19–28.75)	
Current BMI (Kg/m^2^)	17.68 (16.05–19.46)	
Current weight (Kg)	47 (43.1–52.95)	
Current height (cm)	163.57 +/− 5.96	
FFM (Kg)	37.43 +/− 4.3	
FM (Kg)	10.14 +/− 5.41	
AN duration (years)	5 (2–11)	
AN subtype		Restrictive: 66 (47.82%) Non-restrictive: 69 (50%)
Food selectivity		Yes: 100 (72.46%) No: 36 (26.08%)
Meal skipping		Yes: 56 (40.58%) No: 79 (57.24%)
Bulimic episodes		Yes: 46 (33.33%) No: 90 (65.21%)
Vomiting		Yes: 55 (39.85%) No: 81 (58.69%)
Physical hyperactivity		Yes: 66 (47.82%) No: 70 (50.72%)
Snack eating		Yes: 19 (13.76%) No: 117 (84.78%)
Compulsory eating		Yes: 31 (22.46%) No: 103 (74.63%)
Polydipsia		Yes: 10 (7.24%) No: 126 (91.3%)
Fasting practices		Yes: 49 (35.5%) No: 87 (63.04%)
Laxative use		Yes: 16 (11.59%) No: 120 (86.95%)
Nicotine consumption		Yes: 48 (34.78%) No: 90 (65.21%)
Current depressive episode		Yes: 35 (25.36%) No: 88 (63.76%)
Current anxiety disorder		Yes: 65 (47.1%) No: 59 (42.75%)
Current obsessive compulsive disorder		Yes: 12 (8.69%) No: 111 (80.43%)
Current post-traumatic stress disorder		Yes: 5 (3.62%) No: 118 (85.5%)
Current alcohol and/or substance use disorder		Yes: 8 (5.79%) No: 114 (82.6%)
History of suicide attempt		Yes: 28 (20.29%) No: 109 (78.98%)
EDE-Q restriction	3.6 (1.6–4.8)	
EDE-Q eating concern	2.8 (1.6–4)	
EDE-Q weight concern	4 (2–4.8)	
EDE-Q shape concern	4.43 (2.68–5.37)	
EDE-Q total score	3.53 (2.45–4.43)	
d2 score ratio	0.032 (0.017–0.056)	
REE_IC_ (Kcal)	1135.65 +/− 202.2	
REE_HB_ (Kcal)	1296.27 (1242.04–1340.57)	
REE_Sch_ (Kcal)	943.12 (850.37–1031.65)	
REE_Muller1_ (Kcal)	851.48 (768.84–938.25)	
REE_Muller2_ (Kcal)	1087.91 (1010.9–1174.9)	

* If normally distributed. Grey columns correspond to statistics that could not be done because of the variable’s type.

**Table 2 nutrients-14-02727-t002:** Bivariate analysis and multivariate analysis of variables associated with REE_IC._

Variable	Bivariate Analysis with REE_IC_ as Dependent Variable: Statistical Test (Correlation for Continuous Variables and Comparison of Means for Categorical Variables) Mean ± SD for Normally Distributed Data or Median (Q1–Q3)	*p* Value	Linear Regression Model (Model 1) with REE_IC_ as Dependent Variable and All Eligible Independent Variables (Adjusted R^2^ = 0.559; *df* = 119; *p* < 0.001)	Final Linear Regression Model (Model 2) with REE_IC_ as Dependent Variable (Adjusted R^2^ = 0.579; *df* = 123; *p* < 0.001)
Age (years)	Spearman’s Rho = 0.068	0.437		
Current BMI (Kg/m^2^)	Spearman’s Rho = 0.657	**<0.001**		
Current weight (Kg)	Spearman’s Rho = 0.689	**<0.001**		
Current height (cm)	Pearson’s correlation; r = 0.253	**0.004**		
FFM (Kg)	Pearson’s correlation; r = 0.595	**<0.001**	**<0.001**	***p* < 0.001**
FM (Kg)	Spearman’s Rho = 0.599	**<0.001**	**<0.001**	***p* < 0.001**
AN duration (years)	Spearman’s Rho = 0.203	**0.021**	**0.007**	*p* = 0.058
AN subtype	*T* test: AN restrictive type: 1052.4 ± 177.56 AN non-restrictive type: 1205.4 ± 184.96	**<0.001**	0.069	***p* = 0.005**
Food selectivity	*T* test: Yes: 1119.63 ± 207.18 No: 1184.33 ± 184.92	0.115	0.221	*p* = 0.1
Meal skipping	*T* test: Yes: 1168.56 ± 212.28 No: 1105.71 ± 187.06	0.078	0.938	
Bulimic episodes	*T* test: Yes: 1187 ± 173.59 No: 1110.47 ± 212.51	**0.041**	0.586	
Vomiting	*T* test: Yes: 1174.09 ± 186.8 No: 1110.58 ± 210.58	0.081	0.907	
Physical hyperactivity	*T* test: Yes: 1136.71 ± 198.01 No: 1135.7 ± 208.85	0.978		
Snack eating	*T* test: Yes: 1265.16 ± 197.13 No: 1115.27 ± 196.94	**0.003**	0.931	
Compulsory eating	*T* test: Yes: 1217.89 ± 211.27 No: 1108.92 ± 192.67	**0.01**	0.908	
Polydipsia	*T* test: Yes: 1173 ± 210.89 No: 1113.42 ± 203.04	0.575		
Fasting practices	*T* test: Yes: 1167.02 ± 214.33 No: 1117.91 ± 195.04	0.185		
Laxative use	*T* test: Yes: 1174.68 ± 216.24 No: 1130.73 ± 201.48	0.42		
Nicotine consumption	*T* test: Yes: 1203.8 ± 173.03 No: 1098.77 ± 208.13	**0.004**	0.135	*p* = 0.115
Current depressive episode	*T* test: Yes: 1109.57 ± 218.78 No: 1137.65 ± 204.97	0.514		
Current anxiety disorder	*T* test: Yes: 1106.79 ± 233.51 No: 1157.6 ± 172.35	0.184		
Current obsessive compulsive disorder	*T* test: Yes: 1172.83 ± 222.58 No: 1124.91 ± 207.24	0.453		
Current post-traumatic stress disorder	*T* test: Yes: 1225.2 ± 350.58 No: 1125.56 ± 201.34	0.297		
Current alcohol and/or substance use disorder	*T* test: Yes: 1223.57 ± 160.8 No: 1124.56 ± 211	0.226		
History of suicide attempt	*T* test: Yes: 1210.5 ± 218.1 No: 1117.37 ± 193.63	**0.03**	0.911	
EDE-Q restriction	Pearson’s correlation; r = −0.013	0.884		
EDE-Q eating concern	Pearson’s correlation; r = −0.07	0.451		
EDE-Q weight concern	Pearson’s correlation; r = 0.132	0.156		
EDE-Q shape concern	Pearson’s correlation; r = 0.134	0.152		
EDE-Q total score	Pearson’s correlation; r = 0.07	0.462		
d2 score ratio	Pearson’s correlation; r = 0.042	0.667		

Statistically significant *p* values are displayed in bold character. Grey columns correspond to independent variables that were not selected for the linear regression model.

**Table 3 nutrients-14-02727-t003:** Bivariate and multivariate analyses of factors associated with Diff_(Sch-IC)_ variability.

Variable	Bivariate Analysis with Diff_(Sch-IC)_ as Dependent Variable: Statistical Test (Correlation for Continuous Variables and Comparison of Means for Categorical Variables) Mean ± SD for Normally Distributed Data or Median (Q1–Q3)	*p* Value	Linear Regression Model (Model 3) with Diff_(Sch-IC)_ as Dependent Variable and All Eligible Independent Variables Included (Adjusted R^2^ = 0.349; *df* = 119; *p* < 0.001)	Final Linear Regression Model (Model 4) with Diff_(Sch-IC)_ as Dependent Variable (Adjusted R^2^ = 0.381; *df* = 121; *p* < 0.001)
Age (years)	Spearman’s Rho = −0.497	**<0.001**	***p* = 0.001**	***p* < 0.001**
Current BMI (Kg/m^2^)	Spearman’s Rho = −0.143	0.103	*p* = 0.555	
Current weight (Kg)	Spearman’s Rho = −0.026	0.767		
Current height (cm)	Pearson’s correlation; r = 0.232	**0.008**	***p* = 0.009**	***p* = 0.007**
FFM (Kg)	Pearson’s correlation; r = 0.096	0.273		
FM (Kg)	Spearman’s Rho = −0.121	0.168		
AN duration	*T* test: ≤3 years: −77.88 ± 146.39 >3 years: −245.3 ± 149.14	**<0.001**	***p* = 0.025**	***p* = 0.012**
AN subtype	*T* test: AN restrictive type: −136.06 ± 164.73 AN non-restrictive type: −225.25 ± 157.1	**0.002**	*p* = 0.549	*p* = 0.23
Food selectivity	*T* test: Yes: −164.12 ± 163.64 No: −239.71 ± 172.44	**0.026**	*p* = 0.124	*p* = 0.068
Meal skipping	*T* test: Yes: −210.59 ± 173.42 No: −160.62 ± 162.56	0.097	*p* = 0.694	
Bulimic episodes	*T* test: Yes: −166.8 ± 170 No: −216.77 ± 162.39	0.113	*p* = 0.957	
Vomiting	*T* test: Yes: −214.3 ± 158.56 No: −162.36 ± 172.85	0.088	*p* = 0.952	
Physical hyperactivity	*T* test: Yes: −165.99 ± 163.03 No: −199.13 ± 173.01	0.266		
Snack eating	*T* test: Yes: −283.72 ± 162.74 No: −167.2 ± 164.46	**0.006**	*p* = 0.821	
Compulsory eating	*T* test: Yes: −228.31 ± 153.93 No: −167.53 ± 169.29	0.088	*p* = 0.781	*p* = 0.545
Polydipsia	*T* test: Yes: −237.68 ± 215.52 No: −179.39 ± 164.89	0.319		
Fasting practices	*T* test: Yes: −211.65 ± 177.45 No: −166.75 ± 166.83	0.144	*p* = 0.384	*p* = 0.254
Laxative use	*T* test: Yes: −246.34 ± 199.8 No: −174.55 ± 162.7	0.111	*p* = 0.979	
Nicotine consumption	*T* test: Yes: −230.83 ± 148.5 No: −157.49 ± 172.36	**0.016**	*p* = 0.31	*p* = 0.208
Current depressive episode	*T* test: Yes: −158.88 ± 172.69 No: −184.59 ± 170.19	0.465		
Current anxiety disorder	*T* test: Yes: −180.74 ± 182 No: −174.91 ± 156.85	0.853		
Current obsessive compulsive disorder	*T* test: Yes: −206.51 ± 192.88 No: −174.11 ± 168.54	0.535		
Current post-traumatic stress disorder	*T* test: Yes: −142.07 ± 226.39 No: −178.97 ± 168.81	0.638		
Current alcohol and/or substance use disorder	*T* test: Yes: −188.53 ± 129.52 No: −176.6 ± 174.03	0.859		
History of suicide attempt	*T* test: Yes: −231.8 ± 164.29 No: −169.16 ± 167.32	0.081	*p* = 0.593	
EDE-Q restriction	Pearson’s correlation; r = 0.093	0.311		
EDE-Q eating concern	Pearson’s correlation; r = 0.025	0.787		
EDE-Q weight concern	Pearson’s correlation; r= −0.059	0.528		
EDE-Q shape concern	Spearman’s Rho= −0.02	0.832		
EDE-Q total score	Pearson’s correlation; r = 0.018	0.85		
d2 score ratio	Pearson’s correlation; r = −0.091	0.344		

Statistically significant *p* values are displayed in bold character. Grey columns correspond to independent variables that were not selected for the linear regression model.

**Table 4 nutrients-14-02727-t004:** Post-hoc analysis comparing each estimated REE with REE_IC_ after the repeated measures ANOVA.

		Mean Difference	SE	t	p_bonf_
REE_IC_	REE_HB_	−160.511	9.500	−16.896	<0.001
	REE_Sch_	185.138	9.500	19.489	<0.001
	REE_Muller1_	274.368	9.500	28.882	<0.001
	REE_Muller2_	34.158	9.500	3.596	0.005
	Adjusted REE_Sch_	0.150	9.500	0.016	1.000

## Data Availability

The data presented in this study are available on request from the corresponding author. The data are not publicly available due to privacy reasons.

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
