# Peer review of "Clinical Correlates of Measured and Predicted Resting Energy Expenditure in Patients with Anorexia Nervosa: A Retrospective Cohort Study"

_nutrients, 2022, doi:10.3390/nu14132727_

Round 1

Reviewer 1 Report

This is an interesting study that should advance clinical research in eating disorders.

Some editorial comments include the following:

Line 53, this reviewer is not aware of any hospital that uses a Douglas bag, most use portable indirect calorimeters.

Line 65, most hospitals have access to a BIA, even a scale with BIA is frequently used in many hospitals.

General comment, the authors do not offer any biological reason that the other parameters they include in the new prediction model would influence basal metabolism.

Statistical methods: The writing of this section is important, but not easy to follow.

First, the authors should include a detailed description of which variables were included in the backward regression.  Then they should report those variables that remained.  Expecting the reader to go from table to table to see this is disrespectful and weakens the section.

Second, the authors seem to suggest that they included BMI, weight, and height, as well as FFM and FM in the backward regression.  This would be wholly inappropriate as they are simply over controlling for height and weight.  The general rule is that if you include height, FM, and FFM, you never also include BMI and weight.  Please explain more clearly or repeat the analyses.

Discussion: This section will be revised as new results are reported if the statistical analyses were as written.

Author Response

Reviewer 1:

This is an interesting study that should advance clinical research in eating disorders.

Some editorial comments include the following:

Remark 1: Line 53, this reviewer is not aware of any hospital that uses a Douglas bag, most use portable indirect calorimeters.

Response: We thank the reviewer for this remark. We meant that the most accurate method to measure resting energy expenditure requires a Douglas bag, which is not practical to use in hospitals’ settings. We corrected the corresponding sentence and added two references for better clarity:

REE represents the amount of calories required by the body for 24 hours during a non-active period. Indirect calorimetry techniques (e.g. metabolic cart and Douglas bag) are the gold standard method for measuring REE, but they are difficult to apply in standard clinical and hospital settings [7-9]. Several other computerized techniques are more practical to use in such settings, but they may present a variable margin of error due to the technology heterogeneity [8, 9].

Remark 2: Line 65, most hospitals have access to a BIA, even a scale with BIA is frequently used in many hospitals.

Response: We thank the reviewer for this remark. As a matter of fact, we wanted to give a global picture about how FFM and FM are determined. We agree that BIA is available in middle to high income countries. However, this is not universal. To give the reader more information on this subject we added five references and the following paragraph:

“FFM and FM can be determined by bioelectrical impedance analysis (BIA) or dual energy X-ray absorptiometry. BIA relies on formulae that use the measured reactance and resistance to estimate FFM and FM in different population settings [17, 18]. Conversely, dual energy X-ray absorptiometry directly measures the body composition, but it is an expensive technique that sometimes is not available in clinical settings [19]. The level of agreement between BIA and dual energy X-ray absorptiometry findings is debated [20, 21]”.

Remark 3: General comment, the authors do not offer any biological reason that the other parameters they include in the new prediction model would influence basal metabolism.

Response: We thank the reviewer for this important remark. Data from indirect studies suggest that acute AN is different from chronic AN in terms of body composition, appetite hormone effect, impact of hypercortisolism, etc. Accordingly, we added the following information to better explain our results and hypotheses:

“In addition, the impact of the sustained hypercortisolism and appetite hormone dysregulation and the disease effect on adipose tissue and muscle mass are different in acute and chronic AN [34, 35]. Besides these factors, chronic starvation and other behavioral phenotypes, such as physical hyperactivity and induced vomiting, may differentially alter body composition, with consequences on several important clinical, biological and prognostic parameters, including REE [36-38]. However, it is not known whether AN duration influences REE.      ”

Statistical methods: The writing of this section is important, but not easy to follow.

First, the authors should include a detailed description of which variables were included in the backward regression. Then they should report those variables that remained. Expecting the reader to go from table to table to see this is disrespectful and weakens the section.

Response: We thank the reviewer for this important remark. We did not include the list of variables in the original version of the manuscript only because some editors sometimes ask to display the final analysis without the procedure details. We agree with the reviewer that the variables should be precisely mentioned and we apologize if the statistical methods were difficult to follow. We have now better explained the two models of linear regression (model 1 and 2) for REEIC used as a dependent variable, and created a column for model 1 in Table 2. We also mentioned, in the statistical methods section, all variables included in the backward regression and reported the variables that remained as follows:

In Model 1, all eligible independent variables were included (FM, FFM, AN duration in quartiles, AN subtype, food selectivity, meal skipping, compulsory eating, snack eating, bulimic episodes, vomiting, nicotine consumption, and history of suicide attempts). As BMI, height and weight were highly correlated with FM and FFM, they were not included in Model 1. In Model 2, only independent variables with the lowest p value and explaining the REEIC variability remained (FM, FFM, AN duration, AN subtype, food selectivity, and nicotine consumption).   

We also created two models for Diff(Sch-IC) (model 3 and 4) and added one column for model 3 in Table 3 and added the following text:

In Model 3, all eligible independent variables were included (age, current BMI, current height, AN duration in two categories, AN subtype, food selectivity, meal skipping, bulimic episodes, vomiting, snack eating, compulsory eating, fasting, laxative use, nicotine consumption, and history of suicide attempts). As current weight, FFM and FM were highly correlated with BMI and height, they were excluded from the model. In Model 4, only independent variables with the lowest p value and better explaining Diff(Sch-IC) variability remained (age, AN duration, AN subtype, current height, compulsory eating, food selectivity, fasting, nicotine consumption).

Remark 4: Second, the authors seem to suggest that they included BMI, weight, and height, as well as FFM and FM in the backward regression. This would be wholly inappropriate as they are simply over controlling for height and weight. The general rule is that if you include height, FM, and FFM, you never also include BMI and weight.  Please explain more clearly or repeat the analyses.

Response: We thank the reviewer for this remark. We agree that this was not clear enough in the Statistical methods section. In models 1 and 2, FM and FFM were more solid as predictors of REEIC than BMI, height and weight. Accordingly, we included only FFM and FM. In models 3 and 4, BMI and height were more robust as predictors of the difference between REESch and REEIC and therefore, current weight, FM and FFM were not included.

Remark 5: Discussion: This section will be revised as new results are reported if the statistical analyses were as written.

Response: We thank the reviewer for this remark. However, our results were not changed.

Reviewer 2 Report

The author should rewrite the introduction and the beginning of the discussion. Data were not clearly presented.

"My general suggestion is that the authors involve a native speaker to read their paper. Often, sentences are too long and the key message is lost. Line 22: add a short definition of REE Line 29: "Clinical correlates is too vague" Line 47-49: Sentence is too long. Line 72-74: please rephrase In the introduction, please properly define REE. Line 101: Replace "Until now" with "to our best knowledge" Line 105-114: Sentence too long. Line 107: "etc" should be removed. Line 109-114: Please, rewrite in a straight way to convey the message. Sentence is too complicated and long. Line 117-119: Please rewrite

The discussion requires extensive English editing."

Author Response

The author should rewrite the introduction and the beginning of the discussion. Data were not clearly presented.

Remark 1: My general suggestion is that the authors involve a native speaker to read their paper. Often, sentences are too long and the key message is lost.

Response: We used the service of a professional to edit the manuscript for English language. 

Remark 2: Line 22: add a short definition of REE

Response: We thank the reviewer for this remark. We have added the following:

Resting energy expenditure (REE) represents the amount of calories required for a 24-hour period by the body during a non-active period. It is an important parameter in nutrition rehabilitation of patients with anorexia nervosa (AN).

Remark 3: Line 29: "Clinical correlates is too vague"

Response: The reviewer is right about the fact that we need to be more clear about the study objective. We added:

This study determined whether age, body mass index, AN duration/subtype/ specific symptoms/clinical severity, cognitive function alterations, and psychiatric comorbidities influenced REE or the difference between the calculated and estimated REE.

Remark 4: Line 47-49: Sentence is too long. 

Response: We divided the sentence in two:

In the available nutritional rehabilitation programs for AN, the calorie levels prescribed to promote weight restoration are often based on resting energy expenditure (REE) estimates. REE represents the amount of calories required by the body for 24 hours during a non-active period. Indirect calorimetry techniques (e.g. metabolic cart and Douglas bag) are the gold standard method for measuring REE, but they are difficult to apply in standard clinical and hospital settings [7-9]. Several other computerized techniques are more practical to use in such settings, but they may present a variable margin of error due to the technology heterogeneity [8, 9]. Therefore, predictive formulae to calculate REE, based on age, height and body weight, such as the Harris-Benedict equation (elaborated for patients with malnutrition [6, 10]), the Schebendach equation (an adaptation of the Harris-Benedict equation for patients with AN) [11] and the Muller equation [an adaptation of the Harris-Benedict equation for individuals with a body-mass index (BMI) <18.5] are widely used by clinicians [12].

Remark 5: Line 72-74: please rephrase In the introduction, please properly define REE.

Response: We have defined REE in the Introduction as previously mentioned. In addition, the sentence 72-74 was replaced by:

In addition, although many alterations are similar in patients with AN and in patients with malnutrition due to other causes, several AN-specific nutritional aspects, such as some behavior patterns, food selectivity, neuroendocrine changes and clinical comorbidities, may increase the difference between predicted and measured REE [6, 22]. This discrepancy between predicted and measured REE values may partly explain why in some patients with AN, nutritional rehabilitation fails (i.e. lack of weight restoration if the estimated REE is lower than the true REE, or refeeding syndrome if the estimated REE is higher than true REE [23]).

Remark 6: Line 101: Replace "Until now" with "to our best knowledge"

Response: We have replaced “Until now” with “To the best of our knowledge” 

Remark 7: Line 105-114: Sentence too long.

Response: We rephrased this paragraph according to the reviewer’s remarks:

To the best of our knowledge, no study simultaneously assessed the association of several possible AN categories with REE variability. In addition, all the available equations to estimate REE take into account only demographic, anthropometric, and body composition data (age, sex, height, weight, FFM and FM) without considering that some specific AN clinical features also may be important. We hypothesized that some clinical features (e.g. AN duration and/or subtype, comorbid mental disorders, specific accompanying symptoms, cognitive function alterations, and clinical severity) may affect REE. The main objective of this study was to identify AN-related clinical variables and/or comorbidities that affect REE measured by indirect calorimetry and/or are associated with higher differences between predicted and measured REE. The secondary objective was to propose a modified equation for REE estimation that takes into account these significant variables and that is in agreement with the measured REE and easy-to-determine in clinical settings.

Remark 8: Line 107: "etc" should be removed. 

Response: Thank you for this remark. We have rephrased the sentence as follows:

We hypothesized that some clinical features (e.g. AN duration and/or subtype, comorbid mental disorders, specific accompanying symptoms, cognitive function alterations, and clinical severity) may affect REE.

Remark 9: Line 109-114: Please, rewrite in a straight way to convey the message. Sentence is too complicated and long. 

Response: We have now rephrased this paragraph as follows:

To the best of our knowledge, no study simultaneously assessed the association of several possible AN categories with REE variability. In addition, all the available equations to estimate REE take into account only demographic, anthropometric, and body composition data (age, sex, height, weight, FFM and FM) without considering that some specific AN clinical features also may be important. We hypothesized that some clinical features (e.g. AN duration and/or subtype, comorbid mental disorders, specific accompanying symptoms, cognitive function alterations, and clinical severity) may affect REE. The main objective of this study was to identify AN-related clinical variables and/or comorbidities that affect REE measured by indirect calorimetry and/or are associated with higher differences between predicted and measured REE. The secondary objective was to propose a modified equation for REE estimation that takes into account these significant variables and that is in agreement with the measured REE and easy-to-determine in clinical settings.

Remark 10: Line 117-119: Please rewrite

Response: We have rewritten this sentence as follows:

Eligible patients were all consecutive outpatients with AN, according to the Diagnostic and Statistical Manual of Mental Disorders 5th edition (DSM-5) criteria, who were seen at the eating disorder unit of Montpellier, France, between May 2017 and January 2020 [39]. Remark 11: The discussion requires extensive English editing

Response: We have extensively edited the “Discussion” with the help of a professional English language editor.